

# Epidemiological correlates of overweight and obesity in the Northern Cape Province, South Africa

Mackenzie H. Smith[1], Justin W. Myrick[2], Oshiomah Oyageshio[3], Caitlin Uren[4,5], Jamie Saayman[4], Sihaam Boolay[4], Lena van der Westhuizen[2], Cedric Werely[4], Marlo Möller[4,5], Brenna M. Henn[2] and Austin W. Reynolds[1]

[1] Department of Anthropology, Baylor University, Waco, United States
[2] Department of Anthropology and UC Davis Genome Center, University of California, Davis, Davis, United States
[3] Center for Population Biology, University of California, Davis, Davis, United States
[4] DSI-NRF Centre of Excellence for Biomedical Tuberculosis Research, South African Medical Research Council Centre for Tuberculosis Research, Division of Molecular Biology and Human Genetics, Faculty of Medicine and Health Sciences, University of Stellenbosch, Cape Town, South Africa
[5] Centre for Bioinformatics and Computational Biology, University of Stellenbosch, Cape Town, South Africa

Corresponding author
Austin W. Reynolds,
austin_reynolds@baylor.edu

## ABSTRACT

**Background:** In the past several decades, obesity has become a major public health issue worldwide, associated with increased rates of chronic disease and death. Like many developing nations, South Africa is experiencing rapid increases in BMI, and as a result, evidence-based preventive strategies are needed to reduce the increasing burden of overweight and obesity. This study aimed to determine the prevalence and predictors of overweight and obesity among a multi-ethnic cohort from the rural Northern Cape of South Africa.

**Methods:** These data were collected as part of a tuberculosis (TB) case-control study, with 395 healthy control participants included in the final analysis. Overweight and obesity were defined according to WHO classification. Multivariate linear models of BMI were generated using sex, age, education level, smoking, alcohol consumption, and diabetes as predictor variables. We also used multivariable logistic regression analysis to assess the relationship of these factors with overweight and obesity.

**Results:** The average BMI in our study cohort was 25.2. The prevalence of overweight was 18.0% and the prevalence of obesity was 25.0%. We find that female sex, being older, having more years of formal education, having diabetes, and being in a rural area are all positively associated with BMI in our dataset. Women (OR = 5.6, 95% CI [3.3–9.8]), rural individuals (OR = 3.3, 95% CI [1.9–6.0]), older individuals (OR = 1.02, 95% CI [1–1.04]), and those with more years of education (OR = 1.2, 95% CI [1.09–1.32]) were all more likely to be overweight or obese. Alternatively, being a smoker is negatively associated with BMI and decreases one's odds of being overweight or obese (OR = 0.28, 95% CI [0.16–0.46]).

**Conclusions:** We observed a high prevalence of overweight and obesity in this study. The odds of being overweight and obese were higher in women, those living in rural areas, and those with more education, and increases with age. Community-based

interventions to control obesity in this region should pay special attention to these groups.

# INTRODUCTION

Obesity, a medical condition characterized by excessive fat accumulation, can have severe consequences for health, including increased risk of cardiovascular disease, certain cancers, and stroke (*Centers for Disease Control and Prevention, 2021b*). Obesity is most commonly determined through the measurement of Body Mass Index (BMI), a metric calculated by dividing an individual's weight in kilograms by their height in meters squared. From 1990 to 2015, global mortality associated with elevated BMI increased by 28.3%, with the majority of these deaths being caused by cardiovascular disease (*The GBD 2015 Obesity Collaborators, 2017*). In addition to its effects on individual health, high obesity incidence can have substantial economic implications, with a global impact of approximately $2.0 trillion annually (*Woetzel et al., 2014*). The costs linked to obesity range from medical expenses and pharmaceuticals to absenteeism and premature mortality (*Dee et al., 2014*).

Global obesity prevalence increased almost three-fold between 1975 and 2016 (*World Health Organization, 2021*). A 2015 study showed that 603.7 million adults globally were classified as obese, representing 12.0% of the adult population (*The GBD 2015 Obesity Collaborators, 2017*). Based on current trends, it has been estimated that 1.12 billion adults worldwide will be classified as obese by 2030, and an additional 2.16 billion as overweight (*Popkin, Adair & Ng, 2012*), roughly 40% of the world population. Sub-Saharan Africa in particular has shown staggering increases in obesity prevalence in the past several decades. Among seven countries surveyed in this region in 2009, an average of 31.4% of women were classified as overweight or obese (*Ziraba, Fotso & Ochako, 2009*). But this varies by sex, with men frequently having much lower rates of obesity compared with women (*Price et al., 2018*).

The etiology of overweight and obesity is multi-factorial with many genetic and environmental inputs that have been previously identified. Behavior plays an important role, with nutrition, physical activity levels, and sleeping patterns all being previously associated with an individual's BMI (*Neuman et al., 2011*). Smoking is correlated with lower BMI, and BMI tends to increase after individuals stop smoking (*Piirtola et al., 2018*; *Micklesfield et al., 2018*). Alcohol consumption has also been linked to BMI outcomes, however whether this correlation is positive or negative has been widely debated in the literature (*Sayon-Orea, Martinez-Gonzalez & Bes-Rastrollo, 2011*). Understanding the factors involved in overweight and obesity is of high importance to public health researchers, as obesity is a risk factor for type 2 diabetes, among other diseases that are increasing in many LMICs (*Barceló et al., 2007*; *Teufel et al., 2021*).

In accordance with both global trends and those observed in sub-Saharan Africa, the population of South Africa has also seen a rapid increase in average BMI over the past

several years (*Cois & Day, 2015*; *Steyn et al., 1990*; *Steyn et al., 1998*). As of 2016, 33% of men and 68% of women in South Africa were categorized as overweight or obese (*Statistics South Africa, South Africa, editors, 2016*). These values are higher than what has been reported in other African nations (*Goedecke, Jennings & Lambert, 2006*). Various behavioral factors have also been studied in South African populations, with several studies indicating an inverse correlation between smoking and BMI (*Micklesfield et al., 2018*; *Wagner et al., 2018*). The higher obesity incidence among women as compared to men in South Africa is a trend often observed in developing nations (*Popkin, Adair & Ng, 2012*; *Goedecke, Jennings & Lambert, 2006*; *Okop, Levitt & Puoane, 2015*).

It is important to note, however, that the majority of research on BMI and obesity in South Africa has been conducted on urban cohorts, particularly in the Eastern and Western Cape Provinces (*Malhotra et al., 2008*; *Owolabi, Ter Goon & Adeniyi, 2017*). Although national statistics on measures of obesity are available that are representative of both urban and rural communities in South Africa (*Statistics South Africa, South Africa, editors, 2016*), these are reported at the provincial level as part of the South Africa Demographic and Health Survey (DHS). As part of the most recent DHS, only 177 residents of the Northern Cape province were sampled for measures of body mass index (68 men and 109 women) (*Statistics South Africa, South Africa, editors, 2016*). As a result, the variation in overweight and obesity across rural and peri-urban South African districts and municipalities, such as those in the Northern Cape Province, remains limited, restricting our understanding of how overweight and obesity manifest in these communities. This gap in the literature is important given the observed differences in obesity trends between urban and rural communities in lower- and middle-income nations (*Neuman et al., 2013*). In South Africa specifically, women in urban areas were 1.6 times more likely, and urban men 2.3 times more likely, to have excessive BMI than those in rural areas (*Okop, Levitt & Puoane, 2015*). However, the average BMI of both men and women is increasing much more rapidly among rural communities than urban communities globally (*NCD Risk Factor Collaboration (NCD-RisC), 2019*). Further investigation in these smaller communities is crucial in identifying risk factors and working towards improved public health initiatives and education.

In this study, we analyze some of the demographic and behavioral factors related to BMI in the Northern Cape of South Africa. Our sample consists of 395 individuals from rural areas, small towns, and one large municipality in the province. Investigating these factors will give us a more fine-grained understanding of obesity across the region and lay the groundwork for the development of specialized public health and education initiatives to reduce the prevalence of obesity and overweight among this region.

## MATERIALS AND METHODS

### Study design and sampling procedure

The data used in this study were collected as part of the Northern Cape Tuberculosis (NCTB) Project, a case-control study on host susceptibility to TB. Between 2017 and the beginning of the SARS-CoV-2 pandemic in 2020, data was collected on 1,095 individuals ($N_{men} = 544$; $N_{women} = 551$) recruited from 12 community (public) health clinics in the

Northern Cape Province, South Africa that serve populations with high TB rates, mostly in rural towns with populations under 10,000. The Northern Cape Province has the largest area, lowest population size and lowest population density among provinces in South Africa (*Statistics South Africa, 2022*). Data was collected *via* participant interviews, medical histories, saliva samples, and anthropometric measurements. Patients ≥18 years and older evaluated for TB were invited to participate. Participants were partitioned into the case or control group based on a decision tree considering previous TB diagnosis, treatment for TB, and TB test results obtained during the course of the study (O. Oyageshio, 2023, unpublished data). In the present study, only participants with a negative TB result were included due to a symptomatic effect of TB on BMI (*Zhang et al., 2017*). Because TB was much more common among men than women in our study (O. Oyageshio, 2023, unpublished data), our final sample of 395 healthy controls included substantially more women than men ($n_{male} = 145$; $n_{female} = 250$). Community healthcare clinics are the front-line for medical care and triage, and often the sole healthcare facility. The sampling strategy was purposively designed for TB cases and controls not for BMI (*i.e.*, convenience sample), however, this sample retains some heterogeneity in controls as most members of the population seek medical care from the community healthcare clinics and because TB exposure is community-wide (*Andrews et al., 2014*; *Gallant et al., 2010*; *Mahomed, 2013*; *Mahomed et al., 2013b*; *Mahomed et al., 2013a*) anyone meeting the minimum criteria for TB evaluation is tested.

## Ethics and informed consent

Data collected from study participants was approved by the Health Research Ethics Committee of Stellenbosch University (Project number: N11/07/210A), the Institutional Review Board of the University of California, Davis (IRB number: 1749073-1) and the Northern Cape Province Department of Health (NC 2015RP14-469). Participation in this study was voluntary, with the ability to withdraw at any time. Written informed consent was obtained and subsequent medical and demographic questionnaires were conducted in the local language of Afrikaans by trained research assistants from the community. All data was kept confidential with no connections to participant names. Deidentified variables were stored in a secure RedCap database following data collection.

## Demographic and socio-economic factors

Demographic information was collected through interviews and recorded on a data collection sheet. Participants were asked to provide their town of residence, highest level of education achieved, and the ethnic group with which they self-identify.

## Medical history

During interviews, participants self-reported diabetes, HIV, asthma, and TB status. If participants reported having diabetes, they were asked to identify if they were a type 1 or type 2 diabetic. HIV status was self-reported as positive, negative, or unknown. Asthma was recorded as no, yes, or unknown. In addition to asking participants whether they had TB at the time of the study, they were also asked if they were currently taking TB

medication, whether they had TB in the past, and if so, how many TB episodes they had experienced. Following the self-reporting of TB information, TB tests were administered to all participants using GeneXpert, Auramine O Stain, GeneXpert Ultra, SMEAR, or Culture tests.

### Behavioral factors

Alcohol use was evaluated by asking participants if they drink alcohol. If yes, further information was collected regarding the specifics of their alcohol consumption. Participants were asked whether they drink beer, wine, liquor, or ginger beer, and how much of each they consume during the week and weekend. Smoking behavior was categorized as yes or no. If participants answered yes to smoking, they were asked to provide the age at which they began smoking, as well as the average amount they smoke per day.

### Anthropometry

Height was measured using a Charder HM200P stadiometer. Participants were asked to look straight ahead and stand with their heels against the wall while measurements were being taken. One height measurement was recorded for participants in pilot data collection and two measurements were taken upon initiation of the primary study. For individuals with two recorded measurements for height, the average of these values was used. Weight was measured with a Seca 876 digital scale and recorded once. Height and weight measurements were used to calculate BMI for each participant using the equation: BMI = $kg/m^2$. Waist and hip circumference measurements were taken with the guidance of a research assistant using a measuring tape, following WHO measurement guidelines (*World Health Organization, 2008*). Waist-to-hip ratio was calculated by dividing average waist circumference (cm) by average hip circumference (cm). Anthropometric variables were collected by community research assistants trained by the researchers. The collection procedures of all research assistants are evaluated every 3–6 months to ensure continued accuracy of the measurements and consistency across research assistants. Following calculation of BMI, participants were assigned to weight categories based on the guidelines set forth by the Centers for Disease Control and Prevention: those with a BMI <18.5 were classified as "Underweight", 18.5 to <25 as "Healthy," 25 to <30 as "Overweight," and 30 or higher as "Obese" (*Centers for Disease Control and Prevention, 2021a*).

### Quality control

Participants who were TB positive, HIV positive, or both ($N = 642$), were removed from this study due to an expected effect of disease on BMI. All hip and waist measurements supervised by community health care worker #5 were also removed due to errors in measurement technique, where waist and hip circumferences were not being measured at the widest point. BMI could not be calculated for an additional 18 participants due to an absence of height measurements, weight measurements, or both, and these individuals were removed. A total of 395 individuals were included in BMI analyses, 206 of which were recruited during pilot data collection. The addition of WHR measurements and additional

**Table 1 Summary of participant demographics, behavioral factors, anthropometry, and BMI.** Summary statistics are provided for men, women, and the total sample. *WHR only available for 160 participants.

| Variable | Men | Women | Total |
|---|---|---|---|
| Participants (N) | 145 | 250 | 395 |
| Age (mean) | 44.2 (±16.2) | 43.2 (±15.1) | 43.5 (±15.5) |
| Years of education (mean) | 7.9 (±3.6) | 8.5 (±3.1) | 8.3 (±3.3) |
| Smokes | 75.2% ($n = 109$) | 53.6% ($n = 134$) | 61.5% ($n = 243$) |
| Drinks alcohol | 49.7% ($n = 72$) | 33.6% ($n = 84$) | 39.5% ($n = 156$) |
| Height (cm) (mean) | 165.2 (±9.6) | 155.7 (±8.5) | 159.2 (±10.0) |
| Weight (kg) (mean) | 57 (±14.3) | 67.3 (±20.8) | 63.5 (±19.3) |
| BMI (mean) | 21.0 (±5.8) | 27.6 (±8.2) | 25.2 (±8.0) |
| WHR (mean)* | 0.84 (±0.1) | 0.88 (±0.2) | 0.87 (±0.2) |
| Underweight | 39.3% ($n = 57$) | 12.4% ($n = 31$) | 22.3% ($n = 88$) |
| Healthy weight | 42.8% ($n = 62$) | 30.8% ($n = 77$) | 35.2% ($n = 139$) |
| Overweight | 13.1% ($n = 19$) | 20.8% ($n = 52$) | 18.0% ($n = 71$) |
| Obese | 4.8% ($n = 7$) | 36.0% ($n = 90$) | 24.6% ($n = 97$) |
| Diabetes prevalence | 3.4% ($n = 5$) | 8.0% ($n = 20$) | 6.3% ($n = 25$) |
| Large town clinic | 35.9% ($n = 52$) | 31.6% ($n = 79$) | 33.2% ($n = 131$) |
| Small town clinic | 39.3% ($n = 57$) | 28.0% ($n = 70$) | 40.0% ($n = 158$) |
| Rural clinic | 24.8% ($n = 36$) | 40.4% ($n = 101$) | 26.8% ($n = 106$) |

questions about smoking and drinking behavior (types and amounts of alcohol/tobacco consumed daily) were added to the collection protocol after the pilot phase. As a result, these additional behavioral variables were not included in the BMI model, and our sample sizes for the WHR measurements are much lower than that of the overall BMI analysis (Table 1).

## Statistical analyses

Quality control and data analysis were performed in R 4.0.2 and ggplot2 (*Wickham, 2016*) was used for data visualization. Seven covariates (alcohol intake (yes or no), smoking (yes or no), diabetes (yes or no), sex, age, clinic type (large town, small town, rural) and years of education) were entered into a generalized linear model to further characterize their association with BMI. As observed in other studies, BMI was not normally distributed among our sample, therefore we used log-transformed BMI as the outcome variable. Large town, the most common outcome for clinic type, was used as the reference for this variable. Bivariate effect plots were generated between BMI and each individual covariate included in the model using the *effects* R package (*Fox, 2003*; *Fox & Weisberg, 2019*).

We also performed statistical analyses using waist-to-hip ratio (WHR) in place of BMI. A generalized linear model was created using WHR as the outcome variable and included alcohol intake, smoking, diabetes, sex, clinic type, years of education, and age. This model included 161 participants. Our generalized linear models were supplemented by binomial models and the generation of odds ratios using the *questionr* package (*Barnier & Briatte,*
*2022*) to investigate the relationship between individual variables and WHO-established BMI categories (overweight/obese *vs* not).

## RESULTS

### Participant demographics

Our final sample (Table 1) included 395 participants ($n_{male}$ = 145; $n_{female}$ = 250). The mean age was 44 years ±15, with all participants falling between 18 and 86 years of age. Our dataset was collected across twelve study sites, representing rural areas (*n* = 158), small towns (*n* = 106), and one small city (*n* = 131) from the Northern Cape Province, South Africa. The most common self-identified ethnicity among participants was Coloured (87%, people of indigenous Khoe-San, Bantu-speaking African, European, Southeast Asian and East Asian ancestry) (*Swart et al., 2021*), followed by Nama (4%) and Tswana (3%) (Table S1). Approximately 7% of participants reported having no education, 27% reported attending or completing primary school only, and 66% reported attending or completing secondary school.

### Behavioral factors

The majority (62%) of participants were smokers. Among women, 54% reported that they smoked, compared to 75% of men. Only 39% of participants reported consuming some quantity of alcohol. This also differed by sex, with 50% of men and 34% of women in our sample reporting alcohol consumption.

### Anthropometry and BMI

Mean height and weight were 159.2 cm ±10.0 (range: 109.4–184.9 cm) and 63.5 kg ±19.3 (range: 27.0–134.0 kg) respectively. The mean BMI was 25.2 ± 8.0. Nearly a quarter (22.3%) of participants were classified as underweight, 35.2% as healthy, 18.0% as overweight, and 24.6% as obese. The distribution of individuals between these classifications differed substantially between male and female participants, with 36.0% of female participants being placed in the obese category compared to only 4.8% of male participants (Table 1).

### Factors associated with BMI, WHR, and obesity

We fit a generalized linear model to investigate the relationship between BMI and the factors of sex, age, education, smoking, drinking, diabetes status, and clinic type. This model showed significant relationships between BMI and smoking, sex, education level, age, diabetes, and rural clinics (Table 2). Together, these factors explained 27% of the variance in BMI among our cohort, with an F-value of 26.54 ($p < 0.001$). Being female showed a strong positive correlation with BMI, as did age, years of education, and rural clinic type. The relationships between each of these variables and BMI were observed in our bivariate effect plots (Fig. S1), which indicated a 5-point BMI increase for women, a 4-point decrease for smoking, a 6-point increase for having 14 years of education as compared to zero, and a 4-point increase for rural clinic locations as compared to large towns

**Table 2 Model 1 results showing the impact of covariates on BMI outcomes in our sample of 395 participants.**

| Variable | Coefficient | Standard error | *p*-value |
|---|---|---|---|
| (Intercept) | 2.71 | 0.08 | <0.001 |
| Alcohol | 0.02 | 0.03 | 0.56 |
| SexF | 0.21 | 0.03 | <0.001 |
| Age | 0.003 | <0.001 | 0.003 |
| Years of education | 0.02 | 0.004 | <0.001 |
| Smokes | −0.14 | 0.03 | <0.001 |
| Diabetes | 0.17 | 0.05 | <0.001 |
| Clinic type (Rural) | 0.14 | 0.03 | <0.001 |
| Clinic type (Small town) | 0.06 | 0.03 | 0.05 |

**Table 3 Model results showing the impact of covariates on odds of being overweight/obese.**

| Variable | Coefficient | OR | Lower CI (2.5%) | Upper CI (97.5%) | *p*-value |
|---|---|---|---|---|---|
| Alcohol | 0.20 | 1.22 | 0.72 | 2.11 | 0.46 |
| SexF | 1.72 | 5.56 | 3.25 | 9.82 | <0.001 |
| Age | 0.03 | 1.03 | 1.01 | 1.05 | 0.01 |
| Years of education | 0.18 | 1.20 | 1.09 | 1.32 | <0.001 |
| Smokes | −1.29 | 0.28 | 0.16 | 0.46 | <0.001 |
| Diabetes | 1.51 | 4.54 | 1.53 | 15.84 | 0.01 |
| Clinic type (Rural) | 1.19 | 3.29 | 1.85 | 5.97 | <0.001 |
| Clinic type (Small town) | 0.58 | 1.79 | 0.97 | 3.36 | 0.07 |

We find evidence that women in our sample had over five times greater odds of being overweight or obese compared to men (OR = 5.56, 95% CI [3.3–9.8], $p < 0.001$; Table 3). In our dataset, increasing age as well as additional years of school increased the odds of being overweight or obese ($p = 0.01$). Smoking, on the other hand, showed a significant negative correlation with BMI ($p < 0.001$). In fact, smoking decreased the odds of being overweight or obese by nearly 75% in our study population (OR = 0.28, 95% CI [0.16–0.46]). Rural clinic type was associated with 3 times greater odds of being overweight or obese (OR = 3.29, 95% CI [1.85–5.97], $p < 0.001$). Additionally, the odds of being overweight or obese were more than 4 times greater in participants with type 2 diabetes ($p = 0.01$).

We also explored the factors associated with waist-to-hip measurement ratios in our dataset. This model was fit on a subset of the dataset (Table S1; $n = 160$), as waist and hip measurements were not available for all participants. We found a slight positive correlation between WHR and years of education ($p = 0.03$), but together these variables only explain ~1% of the variance in WHR in our dataset with an F-value of 1.15 ($p = 0.33$). No other covariates showed a significant association with WHR.

## DISCUSSION

In this study we analyzed the demographic and socio-behavioral factors related to BMI in an adult cohort from the Northern Cape Province, South Africa. Our results suggest a significant difference in obesity trends between men and women in the study population, with 57% of female participants being overweight or obese compared to only 19% of male participants. This may be partly explained by differences in behavior. In our dataset and elsewhere in the published literature (*Micklesfield et al., 2018*; *Rothman, 2008*; *Ramsay et al., 2018*), smoking has a strong negative correlation with BMI in both men and women, with our results indicating that smoking decreases odds of being overweight or obese by 75%. Nicotine use suppresses appetite and causes an increased resting metabolic rate, resulting in weight loss (*Audrain-McGovern & Benowitz, 2011*). With 25% more men in our dataset reporting smoking compared to women, this may in part explain the sex difference in obesity rates observed in our dataset.

Social factors are also likely important in explaining these sex differences in overweight and obesity, as they influence behavioral traits associated with overweight and obesity. Past research has shown that higher BMI is desirable in many African countries, as it is representative of wealth, health (particularly absence of HIV), and fertility in low-income communities where many lack adequate access to food (*Case & Menendez, 2009*; *Naigaga et al., 2018*; *Okop et al., 2016*; *Puoane, Tsolekile & Steyn, 2010*; *Renzaho, 2004*; *Matoti-Mvalo & Puoane, 2011*). This preference, along with women having control over household food spending (*Case & Menendez, 2009*), may be contributing to nutritional behavior of individuals in our study population as well, leading to increased rates of overweight and obesity in women relative to men (*Wagner et al., 2018*). Socioeconomic status, including educational level, is also positively correlated with BMI and negatively correlated with physical activity for both men and women in South Africa (*Ramsay et al., 2018*; *Alaba & Chola, 2014*; *Wandai et al., 2020*), as well as other low- and middle-income nations (*Ramsay et al., 2018*; *Dinsa et al., 2012*; *Gradidge et al., 2018*; *Micklesfield et al., 2014*). This aligns with our result demonstrating the more education one has, the higher their BMI and may be reflective of better access to resources for those individuals.

To better understand our findings within the wider national context, we compared our results to the 2016 South Africa Demographic and Health Survey (*Statistics South Africa, South Africa, editors, 2016*), which collects representative provincial-level data on several public health metrics. Compared with the DHS data on the Northern Cape province as a whole, we find similar levels of overweight and obesity for women in our study area (57% and 62%, respectively). However, we find substantially fewer overweight and obese men (18% *vs* 32%) and substantially more underweight men in our sample compared to the province as a whole (39% *vs* 19%). Interestingly, our findings demonstrate greater odds of being overweight or obese in rural regions than in large towns. While previous data in South Africa has found that men and women in urban areas were more likely to have excessive BMI than those in rural areas, on a global scale, the average BMI of both men and women is increasing much more rapidly among rural communities than urban communities (*NCD Risk Factor Collaboration (NCD-RisC), 2019*). In contrast to our

findings, province-wide measurements from the DHS show no substantial differences in BMI between urban and rural samples, demonstrating the regional variability in obesity and reinforcing the need to collect more data at the district and municipality level to better understand the demographic and social factors that may be influencing obesity on a local scale.

The present study is not without limitations. BMI is a simple and widely used metric for obesity, however, it does not distinguish between excess fat, muscle, or bone mass, or provide any indication of the distribution of fat among individuals (*Rothman, 2008*; *Adab, Pallan & Whincup, 2018*), whereas WHR is better at predicting visceral adiposity and cardiometabolic disease risk (*Song et al., 2013*; *Lee et al., 2008*). Future work incorporating these and other more sensitive measures of adiposity will provide greater detail into the risk-factors and consequences of obesity in this population.

In addition, the sampling strategy for this study was purposively designed to maximize TB cases and controls, not necessarily for maximizing heterogeneity of BMI in the population. Despite this limitation, our controls are fairly evenly distributed across important variables such as age, education, height, and weight. Additionally, because TB exposure is community-wide, reaching ~90% by age 30 (*Andrews et al., 2014*; *Gallant et al., 2010*; *Mahomed, 2013*; *Mahomed et al., 2013b*; *Mahomed et al., 2013a*), TB nurses and healthcare staff test anyone meeting the minimum criteria for TB evaluation; and because these community healthcare clinics are typically the only accessible facility, it is seen by all members of the community. While this sample includes more heterogeneity than a typical convenience sample, it does not fully assuage biases, such as higher SES patients who opt for out-of-area private medical clinics or selection biases in who decided to seek medical attention, among others.

## CONCLUSIONS

Overall, our results suggest that smoking, sex, education level, age, location, and diabetes all influence BMI outcomes among the rural population of the Northern Cape of South Africa, consistent with previous research in other South African cohorts and developing nations. These results deepen our understanding of the factors contributing to obesity risk in this region. They also highlight the regional diversity in BMI across South Africa and the need for more monitoring at the local level. The present study was limited somewhat by the behavioral data collected as part of the larger TB study. A necessary future direction of this work will be to include additional measures of adiposity, physical activity and dietary behaviors, and socio-economic status to obtain a more complete understanding of the risk-factors for obesity in the Northern Cape. This will allow for better identification of high-risk groups for behavioral interventions, ultimately mitigating the public health and economic burdens created by the global obesity epidemic at the local level.

## ACKNOWLEDGEMENTS

First and foremost, we would like to thank our participant communities in the Northern Cape for their continued trust and support in helping us undertake this project. We would also like to thank our community research assistants and translators who assisted in data

collection for the project. Finally, we want to thank the Department of Health in the Northern Cape Province, South Africa for their continued support of the project.

### Funding

This work was funded by NIH grant 5R35GM133531-03 to Brenna M. Henn. The funders had no role in study design, data collection and analysis, decision to publish, or preparation of the manuscript.

### Grant Disclosures

The following grant information was disclosed by the authors:
 NIH grant: 5R35GM133531-03.

### Competing Interests

The authors declare that they have no competing interests.

### Author Contributions

- Mackenzie H. Smith conceived and designed the experiments, analyzed the data, prepared figures and/or tables, authored or reviewed drafts of the article, and approved the final draft.
- Justin W. Myrick performed the experiments, analyzed the data, authored or reviewed drafts of the article, and approved the final draft.
- Oshiomah Oyageshio analyzed the data, authored or reviewed drafts of the article, and approved the final draft.
- Caitlin Uren performed the experiments, authored or reviewed drafts of the article, and approved the final draft.
- Jamie Saayman performed the experiments, authored or reviewed drafts of the article, and approved the final draft.
- Sihaam Boolay performed the experiments, authored or reviewed drafts of the article, and approved the final draft.
- Lena van der Westhuizen performed the experiments, authored or reviewed drafts of the article, and approved the final draft.
- Cedric Werely performed the experiments, authored or reviewed drafts of the article, and approved the final draft.
- Marlo Möller performed the experiments, authored or reviewed drafts of the article, and approved the final draft.
- Brenna M. Henn conceived and designed the experiments, performed the experiments, authored or reviewed drafts of the article, and approved the final draft.
- Austin W. Reynolds conceived and designed the experiments, performed the experiments, analyzed the data, prepared figures and/or tables, authored or reviewed drafts of the article, and approved the final draft.

## Human Ethics

The following information was supplied relating to ethical approvals (*i.e.*, approving body and any reference numbers):

Data collected from study participants was approved by the Health Research Ethics Committee of Stellenbosch University (Project number: N11/07/210A) and the Institutional Review Board of the University of California, Davis (IRB number: 1749073-1).

## Data Availability

The data is available at OSF: Reynolds, Austin W. 2023. "Epidemiological Correlates of Overweight and Obesity in the Northern Cape Province, South Africa." OSF. January 17. osf.io/695ma.

The code used for analysis is available at GitHub: https://github.com/reynolds-lab/NCTB_BMI_Analysis and archived at Zenodo:

awreynolds. (2022). reynolds-lab/NCTB_BMI_Analysis: Publication release (code). Zenodo. DOI 10.5281/zenodo.7245580

## Supplemental Information

Supplemental information for this article can be found online at http://dx.doi.org/10.7717/peerj.14723#supplemental-information.

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
