# Peer review of "Epidemiological correlates of overweight and obesity in the Northern Cape Province, South Africa"

_PeerJ, doi:10.7717/peerj.14723_

## Round 0.1 · original submission · Minor Revisions

This article addresses the prevalence and predictors of obesity in rural areas in South Africa. Overall, the study design is appropriate; however, there are some issues that need to be clarified before we can judge the validity of the study. Please refer to the reviewer’s comments. Below are my comments to move forward with a final decision; some comments coincide with the reviewer’s comments.

Introduction:
The authors have established the literature gap and the significance of the study research question. Please pay attention to the comments of the reviewer regarding updating the references regarding what we know about the topic in South Africa and rural areas. Moreover, do you have more updated references to the national statistics (Reference 19)?
Methods:
Overall the design is appropriate. Nevertheless, some clarifications are needed to assess the validity of the results:
1- Sample size calculation. It seems that the authors have used a convenience sample. This should be mentioned in the methods and highlighted in the limitations sections as it has implications for the generalizability of the results.
2- How were the predictors chosen? Did you have a theoretical or conceptual framework? Did you just rely on the available variables collected by the TB study? This should be clarified in the methods section and limitations sections.
3- Do you have an explanation for why the final sample was not equally distributed between males and females? The initial population of 1,095 individuals was equally distributed. (Men= 544; Women = 551).
4- The reviewer's comment regarding the participants' measuring the hip and waist is very critical. Please clarify. This raises questions about whether the height and weight were measured objectively by the research assistants. Please reconsider whether you want to keep the data on wait and hip circumferences. Even the wording of Lines 165-167 is not clear.
Results and analysis
1- Figure 1 is not needed. Figure 2 can be added as a supplement. There is no need to put the results in the footnote of the tables. This should be mentioned in the text.
2- Please refer to PeerJ - About - Journal Policies & Procedures regarding statistical reporting. Means are lacking standard deviations. You do not have a decimal point for the proportions. P- value is not reported properly. Please be consistent with the use of median vs mean throughout the text and tables. Medians are used if you have a major deviation from the normal distribution.
3- There is a standardized way to report general linear models in the text and tables. Table 3 is not needed as you have mentioned the odds in the next table. It can be replaced by results on the model, its F value and significance, and an explanation of the variance. Be careful of the text explaining the ODDS as you used the log transformation.
3- Regarding the effect of gender. the way you described the analysis is confusing. Please coordinate with a statistician and consider adjusted models or moderation analysis.

·

Basic reporting

The English used throughout is clear and unambiguous. The literature references are sufficient, with the background/context. The article structure is good and the figures and tables are valid. The results are relevant to the study and the data collected.

The article is up to standard.

Experimental design

The research questions are well defined, relevant and meaningful and the research fills the knowledge gap identified in the study. Methods described and used were sufficient and the study can be replicated.

Validity of the findings

This study has a meaningful replication and the rationale and literature reviewed are clearly stated in this manuscript.

The data used by the authors are robust for analysis and the outcome are statistically sound and controlled. Conclusions are well stated and they are stemmed from the findings of the study.

There are no comments to add to the study.

Additional comments

The manuscript has good potential and can be accepted after the author(s) have the paper edited by Professional English Editor

·

Basic reporting

Review
Introduction:
Line 4: certain cancers not all
Line 75: I suggest using a later publication than 2002. Would also suggest using the 2016 SADHS results which provide the N Cape data on overweight and obesity.
Lines 83 -85. I do not agree that rural data is limited. The SADHS provides urban and rural data on overweight and obesity for each province in South Africa for males and females and all ages. I suggest that the authors include the results of the DHS with their own findings. They also give data for different population groups.
Line 94: I think that the behavioural factors such as alcohol use and smoking should also be mentioned as risk factors. Also there is no mention of the effect of obesity on type 2 diabetes.
Line 96: is the large municipality rural? The introduction contains much data on urban and rural differences, yet this is not included as a risk factor for obesity.
Methods
line 107-8: reference required
Line 143-146: detail was asked about smoking and alcohol use but in the tables participants were only classified as smoking or taking alcohol whether a great deal or only occasionally was not included.
Anthropometry: this section needs a reference for the methods used.
Please mention the amount of clothing worn by the participants.
Please provide the make and models of the scales and anthropometers used.
Provide information on who the field workers were and who trained them.
I have never heard of subjects doing their own waist and hip measurements. Please explain how each of these measurements were done. I am not surprised that some were not reliable.
Line 186: was the term questionr correct?
Results:
The term coloured is generally described as Afr-Euro-Malay because of their ancestry.
While 85% were “coloured” and 5% Tswana, what were the other 10%?
Could Type 2 diabetes be added to table 1.
It is usual to please the title above the table and any summaries below it.
In the description of Table 3 it would be important to mention the fact that the odds of overweight/obesity were four times higher with type 2 diabetes.
Discussion
Line 263: I agree with this statement if done by experts and not by the participants
Line 273-4: A number of authors have described the fact that African beauty is determined by having a fuller body shape. It also illustrates that the person does not have HIV or some other disease. Puoane et al. have done some work in this regard.
Another limitation would be the lack of data on physical activity which is also a major risk factor for obesity.
Although the authors have mentioned differences in urban and rural areas regarding obesity in the introduction its is not mentioned in the results. This may be because the sample sizes may have been too small to divide them into geographic areas.
Summary: I would recommend publication if my comments could be addressed.

Experimental design

Aim is well defined.
As mentioned above I question participants measuring their own waist and hip measurements accurately.
Methods secribed reasonably well.

Validity of the findings

The data appears to be robust and statistically sound.
Conclusion is well stated.

Additional comments

All comments f or improvement are in 1. Basic reporting

---

## Round 0.2 · Minor Revisions

I would like to thank you for the modifications and you have answered most of the comments. However, the manuscript still needs more modifications to be fir to be published. I have add a pdf with some comments. Please feel free to accept or modify.

My main points for the minor revision are:

1- more Language editing of the sentences so that paragraphs flow smoothly with storytelling and synthesis rather than just stating facts.

2- more concentration on the data formatting and statistical model. Please refer to scientific guidelines on models, adjusted models and how to present the results.

---

## Round 0.3 · Minor Revisions

You have addressed most of the concerns. I have still two minor comments. Please update the CI to indicate that it is 95% CI.

Regarding the statement "In our dataset, every one-year increase in age is associated with a 2% increase in the odds of being overweight or obese, while each additional year of school completed increased those odds by 20% (p = 0.01). I would be cautious in the narrative as you have used log transformation. Every 2% increase increased the odds of the log-transformed BMI.

---

## Round 0.4 · accepted · Accept

I thank the authors for their patience. The authors have addressed all of the reviewers' and editors' comments.